# Effects of Vaccination against Recombinant FSH or LH Receptor Subunits on Gonadal Development and Functioning Male Rats

**DOI:** 10.3390/vetsci11040176

**Published:** 2024-04-15

**Authors:** Fuqiang Pan, Wanzhen Fu, Bochao Zhang, Mengdi Han, Huihui Xie, Qing Yi, Wei Qian, Jiankun Cui, Meng Cao, Yanqiuhong Li, Yuke Jia, Fugui Fang, Yinghui Ling, Yunsheng Li, Ya Liu

**Affiliations:** 1Departmet of Veterinary Medicine, College of Animal Science and Technology, Anhui Agricultural University, 130 Changjiang West Road, Hefei 230036, China; 17356582644@163.com (F.P.); fwz@stu.ahau.edu.cn (W.F.); 15155931671@163.com (B.Z.); hanmengdi4017@163.com (M.H.); xhui2013@163.com (H.X.); 18855420135@163.com (Q.Y.); qw2230795053@gmail.com (W.Q.); pecodr1114@163.com (J.C.); 16632238281@163.com (M.C.); 18380512787@163.com (Y.L.); 13389161914@163.com (Y.J.); fgfang@ahau.edu.cn (F.F.); lingyinghui@ahau.edu.cn (Y.L.); lys@ahau.edu.cn (Y.L.); 2Anhui Provinciale Key Laboratory of Local Livestock and Poultry, Genetical Resource Conservation and Breeding, College of Animal Science and Technology, Anhui Agricultural University, Hefei 230036, China; 3Linquan County Modern Agriculture Technology Cooperation and Extension Service Center, Fuyang 236000, China

**Keywords:** LHR, FSHR, subunit vaccine, immune castration, male rat

## Abstract

**Simple Summary:**

Castration benefits the management of male livestock and improves their meat quality. Immunocastration is more convenient and aligns with animal welfare, compared to surgical castration. In this study, two subunit vaccines were designed by fusing the conserved antigenic epitopes of porcine FSHR and LHR with the T-helper epitope region of the diphtheria toxin (DTT) and validated in male Sprague Dawley rats. The results showed that both vaccines induced antibody production in rats to some extent, inhibited testicular development, and reduced testicular function.

**Abstract:**

Luteinizing hormone (LH) and follicle-stimulating hormone (FSH) play key roles in regulating testosterone secretion and spermatogenesis in male mammals, respectively, and they maintain the fertility of male animals by binding to their corresponding receptors. We designed and prepared a recombinant LH receptor (LHR) subunit vaccine and a recombinant FSH receptor (FSHR) subunit vaccine and used male Sprague Dawley (SD) rats as a model to examine their effects on testicular development, spermatogenesis, and testosterone secretion in prepubertal and pubertal mammals. Both vaccines (LHR-DTT and FSHR-DTT) significantly decreased the serum testosterone level in prepubertal rats (*p* < 0.05) but had no effect on the testosterone secretion in pubertal rats; both vaccines decreased the number of cell layers in the seminiferous tubules and reduced spermatogenesis in prepubertal and pubertal rats. Subunit vaccine FSHR-DTT decreased the sperm density in the epididymis in both prepubertal and pubertal rats (*p* < 0.01) and lowered testicular index and sperm motility in pubertal rats (*p* < 0.05), whereas LHR-DTT only reduced the sperm density in the epididymis in pubertal rats (*p* < 0.05). These results indicate that the FSHR subunit vaccine may be a promising approach for immunocastration, but it still needs improvements in effectiveness.

## 1. Implications

Immunocastration is a new method of castration that can effectively improve animal welfare and avoid the pain of castration. However, the development of commercial immunocastration vaccines still needs further improvement. This experiment provides two potential immunocastration vaccines and validates them on male rats. We believe that the results of this experiment can contribute to the advancement of immunocastration research and the promotion of immunocastration vaccines.

## 2. Introduction

Castration is usually practiced to reduce the aggressive and crawling behavior of sires [1] and improve the carcass traits of meat livestock [2].Traditionally, castration is achieved through the surgical approach. However, surgical castration frequently causes great stress and impairs animals’ welfare [3]. In past decades, immunocastration has been proved to be an alternative to surgical castration. In addition, immnocastration can also replace lethal means to control the wildlife population and the increasing amount of wandering animals [4]. So far, most castration vaccines are developed based on recombinant or synthetic gonadotrophin-releasing hormones (GnRH), and at least three commercially available vaccines have been launched [5,6,7].

It has been confirmed that there are two forms of GnRH in mammals. The GnRH I, which is usually referred to as GnRH, regulates gonadotropin secretion. GnRH II appears to be a neuromodulator and is found in various peripheral tissues [8], and its physiological role is not determined [1]. Since GnRH II varies from GnRH by only three amino acid residues [8], the antibodies produced by GnRH immunization may cross-react with GnRH II and thus potentially cause unknown harm to the immunized animals. Therefore, it is necessary to develop new castration vaccines as alternatives.

Keeping in view that the follicle-stimulating hormone (FSH) and luteinizing hormone (LH) play crucial roles in regulating the development and function of testes, FSH and LH have been investigated as candidates for developing contraceptive vaccines. When immunized with ovine FSH (oFSH), the spermatogenesis of adult rhesus monkeys was inhibited, and their testes atrophied [9]. Other groups also independently confirmed that oFSH immunization could reduce the fertility in Bonnet monkeys and men [10,11,12]. When vaccinated against purified LH, the serum testosterone and the testicular growth were suppressed in rams and bulls [13]. Since FSH, LH, and thyroid-stimulating hormone (TSH) share the same α subunit [14], antibodies produced by vaccination against FSH or LH react with TSH [15,16]. In order to avoid cross-immunity, the recombinant β subunits of LH (β-LH) [17,18] and FSH (β-FSH) [19,20] were developed as immunogens. Though the β-FSH vaccine has higher targeting specificity, the binding affinity of β-FSH antibody to FSH is lower by one order of magnitude compared to antibodies generated against the intact FSH [21].

Both FSH and LH function by binding to the corresponding receptors. Unlike the continuous production of LH and FSH by the pituitary, the turnovers of the LH receptor (LHR) and the FSH receptor (FSHR) are slow, so it was speculated that antibody titers in mammals immunized with LHR and FSHR vaccines may be maintained at a higher level and last longer than those immunized with gonadotropins or recombinant β subunits [1]. Several recombinant FSHR [22,23,24] and LHR [23,24,25,26] vaccines have been developed and effectively reduced fertility. However, research on the vaccine based on gonadotropins receptors is very limited, and more studies should be conducted to improve their efficacy.

Diphtheria toxin (DT) is a potent exotoxin produced by Corynebacterium diphtheriae, which has strong immunogenicity. The 202–378 portion of its transmembrane domain contains two major T cell epitopes [27] and is capable of eliciting T cell responses. The fusion of the diphtheria toxin transmembrane structure with other proteins will not affect their folding and function [28], and the exogenous epitope transplanted to the diphtheria toxin T domain (DTT) can induce antibody production [29,30,31]. Therefore, we believe that DTT is a suitable immune-enhancing protein. In this study, we developed new recombinant FSHR and LHR subunit vaccines by cloning epitopes of FSHR/LHR into the pET28a-DTT plasmid for fusion expression, and both vaccines were tested in prepubertal and pubertal rats.

## 3. Material and Methods

### 3.1. Experimental Animals

Healthy male Sprague Dawley (SD) rats of the same grandparents were fed in the animal house of Anhui Agricultural University, with free access to water and food, artificial temperature control of 23~26 °C, and natural light. All animal experiments were carried out in accordance with the Regulations of the People’s Republic of China on the Management of Scientific and Technological Laboratory Animals and were approved by the Animal Care and Utilization Committee of Anhui Agricultural University (No. AHAU20208025).

### 3.2. Vaccine Design and Preparation

Porcine FSHR (GI: NP_999551.2) and LHR (GI: NP_999614.1) sequences were searched on Genbank (https://www.ncbi.nlm.nih.gov/) (accessed on 1 April 2020). DNA Star Lasergene (version 7.1.0) software was used to predict the conserved sequences (Appendix A) with good hydrophilicity, flexibility, surface accessibility, and a high antigen index in the extracellular region near the N-terminus of FSHR and LHR. Then, these epitopes were connected in tandem with GGGS as the linker, repeated into the trimer, and cloned into the pET28a-DTT plasmid to construct pET28a-FSHR/LHR-DTT (Appendix A). The reconstructed plasmid was transformed into *E. coli* BL21 (DE3) for expressing the fusion protein [32]. The induction conditions were 1.0 mM IPTG and 26 °C for 8 h. The purified protein was lyophilized with a low-temperature lyophilizer (FD8-3, GOLD-SIM) for 24 h and stored at −80 °C. For vaccine preparation, the lyophilized protein was dissolved with normal saline and mixed with white oil adjuvant (U51021, SEPPIC, Paris, France) at 1:1 (*v*/*v*) and then fully emulsified.

### 3.3. Vaccination and Samples Collection

Prepubertal (20 days old, *n* = 30) and pubertal (6~8 weeks old, *n* = 30) male rats from the same paternal lineage were randomly allocated into 3 groups, respectively, with 10 rats in each group. Rats were injected with 100 μg DTT (control group), LHR-DTT (LHR group), or FSHR-DTT (FSHR group), each into the quadriceps femoris. Blood samples were collected from the infraorbital venous plexus at 0, 2, 4, 6, 9, 12, and 15 weeks after the first immunization, and the serum antibody and testosterone levels were determined; 15 weeks after the first immunization, the weight, transverse diameter, and longitudinal diameter of the testes and the density, viability, and motility of sperm in the epididymis (the above data are the average of the left and right sides) were detected, and the testicular tissue structure was observed.

### 3.4. Antibody Titer Assays

Sequences encoding the reconstructed LHR and FSHR subunits were subcloned from pET-28a-LHR-DTT/pET28a-FSHR-DTT into the pGEX-6P-1 vector, respectively (Appendix A), and the reconstructed plasmids were transformed into BL21 (DE3) *E. coli* for expressing the fusion proteins [33]. The purified GST-LHR and GST-FSHR proteins were diluted to 2 μg/mL with PBS, and 100 μL of the diluted protein solution was added to each well of the 96-well plate for coating overnight at 4 °C. The next day, the coating protein solution was discarded, and the plate was washed with PBST, 200 μL/well, 3 times. Then, the plate was sealed with sealing solution at 37 °C for 2 h. The gradient-diluted serum was added into the microplate in turn, 100 μL/well, and then the plate was incubated at 37 °C for 40 min. The plate was washed three times. The diluted (1:10,000) rabbit anti-rat IgG HL (HRP) (550064, Chengdu Zhengneng Biological Co., Ltd., Chengdu, China) was added into the microplate, 100 μL/well, and the plate was incubated at 37 °C for 40 min. The plate was washed three times. TMB chromogenic solution (1001, Beijing Meikewande Biological Co., Ltd., Beijing, China) was added in the microplate, 100 μL/well, and then the plate was incubated at 37 °C for 15 min in the dark. Reaction termination solution was added into the microplate, 50 μL/well. The optical density was determined by measuring the absorbance at 450 nm using a microplate reader (BioTek, Winooski, VT, USA). The maximum dilution factor at which the OD450 value in the sample well was greater than or equal to twice the OD450 value of the blank control well was defined as the antibody titer.

### 3.5. Testosterone Detection

Testosterone (T) levels were quantified using chemiluminescence immunoassay (CLIA) (MINDRAY, Shenzhen, China), as described previously [34]. Briefly, the method employed in this study is a competitive immunoassay following CLIA principles. Specifically, 50 microliters of serum were loaded onto a reagent cartridge designed for the testosterone assay. The analysis was conducted utilizing the Beckman Kurt UnicelDxl 800 immunoassay system (Beckman Coulter Inc., Brea, CA, USA). In this assay, samples were initially mixed with antibodies labeled with testosterone alkaline phosphatase (ALP) and magnetic latex reagents. Subsequently, a chemiluminescent substrate, 3-[2-spiroadamantane]-4-methoxy-4-[3-phosphoryloxy]-Phenyl-1,2-dioxetane (AMPPD), was introduced to generate chemiluminescent signals, which were quantified by counting the emitted photons. The minimum detection limit was 0.1 ng/mL, with a detection range from 0.1 ng/mL to 16.0 ng/mL and coefficient of variation (CV) values within the range of 5.82% for intra-assay and 4.22% for inter-assay, respectively. The correlation coefficient of the standard curve was 0.9900.

### 3.6. Sperm Density, Viability, and Motility

Fifteen weeks after the first immunization, all rats were euthanized by intraperitoneal injection of excessive chloral hydrate, and the testes and epididymides were quickly collected. The bilateral cauda epididymides were then transferred to a petri dish containing 10 mL of PBS (37 °C, pH 7.2) and cut into pieces with a scalpel blade to release the sperm fully. After incubating for 1 min at 37 °C, 10 μL of sperm suspension was taken along the upper edge of the petri dish and added to the sperm-counting plate. Sperm density, variability, and motility were detected using a sperm analyzer (MD02108, Nanning Songjingtianlun Biotechnology Co., Ltd., Nanning, China).

### 3.7. Histomorphological Observation of Testes

As we described before [35], testes were collected and fixed in 4% paraformaldehyde; then, they were dehydrated in ethanol and embedded in paraffin wax, sectioned, and stained with hematoxylin and eosin [36]. The tissue structure was observed under a microscope (BX51). Images were taken (Nikon Digital Sight DS-SMC camera) (Tokyo, Japan) and saved.

### 3.8. Statistical Analysis

All statistical analyses were performed using IBM SPSS Statistics 24 software, and significance analysis was performed using one-way ANOVA (comparison between three or more groups), independent *t*-test (comparison between two groups), and one-way repeated measures ANOVA (analyzing the relationship between antibodies and time). The results were shown as the mean ± SEM, with *p* < 0.05 indicating differences, *p* < 0.01 indicating significant difference, and *p* > 0.05 indicating no differences. The results were plotted with Graphpad Prism software 3.

## 4. Results

### 4.1. Effect on Body Weight of Rats

To test the effect of immunization against FSHR or LFR on the body weight gain, the rats’ body weights (BW) were monitored during the experiment. As shown in Figure 1A, LHR and FSHR immunizations significantly promoted BW gain in rats at 2 w after initial immunization (*p* < 0.01) and then inhibited the rats’ BW gain dramatically at 9 w post initial immunization (*p* < 0.05). The body weight in the LHR group was significantly lower than that in the control group at 15 w (*p* < 0.01).

Unlike the prepubertal rats, LHR immunization noticeable promoted the BW gain in pubertal rats at 2 w, 9 w, and 12 w after the first immunization (*p* < 0.01), while FSHR immunization only promoted BW gain at 2 w (*p* < 0.05) (Figure 1B).

### 4.2. Antibody Levels

To determine whether the recombinant FSHR and LHR subunit vaccines stimulated an immunoreaction in rats successfully, we detected the corresponding antibodies. As shown in Figure 2, both vaccines induced the production of corresponding antibodies in the prepubertal and the pubertal rats. The antibody levels in all groups rapidly increased to their peak after strengthening immunity and gradually decreased over time. The antibody levels reached the peak at the 9th week in prepubertal rats and the 6th week in pubertal rats, respectively. LHR stimulated a higher antibody titer in prepubertal rats, and the peak titer was 40,500 (Figure 2A,C), while FSHR inducted a higher antibody titer in adult rats, and the peak titer was 135,000 (Figure 2B,D).

### 4.3. Testosterone Levels

As shown in Figure 3A,B, the testosterone levels in the LHR and the FSHR groups were significantly lower than that in the control group in the prepubertal rats 6 weeks after the first vaccination, and this trend persisted until 15 weeks after the primary immunization. However, neither the FSHR nor the LHR vaccine inhibited the secretion of testosterone in pubertal rats (Figure 3C,D).

### 4.4. Testicular Changes

Within the prepubertal rats, there was no significant difference in the testicular index and transverse and longitudinal diameters among the groups 15 weeks after the initial immunization (*p* > 0.05) (Figure 4A–C). In the control group, the seminiferous tubal wall was lined with multiple layers of spermatogenic cells at different development stages, and lumens were filled with a large number of sperm. Contrasted with the control group, the number of cell layers of the tube walls and the number of sperm in the lumens remarkably decreased in the LHR group and the FSHR group (Figure 5).

Unlike the prepubertal rat experiment, FSHR immunization significantly reduced the testis index (*p* < 0.05), while LHR immunization did not affect the size and the index of testes in pubertal rats (*p* > 0.05) (Figure 4D–F). Sections showed that the seminiferous tubules in the control group developed normally, the differentiation of spermatogenic cells was obvious, the cells were closely arranged, and a large number of sperm was formed in the lumen. In the testicular seminiferous tubules of rats immunized with FSHR or LHR, there were remarkable reductions in both spermatogenic cells and sperm (Figure 6).

### 4.5. Sperm Density, Viability, and Motility

Fifteen weeks after the initial immunization, the cauda epididymides of all rats were separated, and the sperm density, viability, and motility were evaluated. In the prepubertal rats, the sperm density in the FSHR group was significantly lower than that in the control group (*p* < 0.01) (Figure 7A). Neither sperm viability nor sperm motility were significantly decreased in both the LHR and the FSHR groups (*p* > 0.05) (Figure 7B,C). In pubertal rats, both FSHR and LHR immunizations reduced the sperm density (*p* < 0.01, *p* < 0.05, respectively). The viability and the motility of sperm in the FSHR group were lower than in those of LHR group (*p* < 0.05) (Figure 7D–F).

## 5. Discussion

In this study, recombinant FSHR and LHR subunit vaccines were prepared, and their efficacies were tested in prepubertal and pubertal male rats.

Although the trend of antibody production in pubertal rats was similar to that in prepubertal rats, the peak of the antibody in pubertal rats appeared 3 weeks earlier than that in prepubertal rats. It is speculated that the production of specific antibodies involves the cooperation and interaction of antigen-presenting cells, T helper cells, and B cells, as well as cytokines [37]. The thymus of a rat is not fully developed at 3 to 4 weeks after birth, the spleen does not show full histological maturity until 6 weeks of age, and the thymus and spleen do not show mature T and B cell populations until 9 weeks of age [38]. In addition, although the B cell population in the spleen of the prepubertal rat is equivalent to that of the pubertal rat, the overall T cell population and T cell subsets are smaller, and the intensity of the specific immune response produced by the body is weaker, so theoretically, the immune response of the pubertal rat is stronger than that of the prepubertal rat and results in a higher antibody titer. Therefore, the peaks of FSHR and LHR antibodies in pubertal rats appeared 3 weeks earlier than that in prepubertal rats. The reason why the titer of LHR antibodies is lower in pubertal rats compared to prepubertal rats is currently unclear.

LH can promote the synthesis and release of testosterone in Leydig cells [39], and testosterone is essential for maintaining spermatogenesis [40]. In this study, due to the high titer of antibodies produced by LHR immunization, the secretion of testosterone was inhibited in prepubertal rats, and correspondingly, the number of sperm in the seminiferous tubules and the sperm density in the cauda epididymidis decreased significantly. In contrast to the prepubertal rats, the serum testosterone levels and the sperm density in the cauda epididymidis did not decrease in pubertal rats, due to a low level of LHR antibodies.

In this study, FSHR vaccination significantly reduced the serum testosterone level of prepubertal rats, resulting in significantly sparse spermatozoa in the seminiferous tubules and significantly lower sperm density in the epididymis, compared with the control group. These results are consistent with those in mice [41,42]. This is because FSH plays an important role in spermatogenesis [43,44,45] and directly stimulates Sertoli cell proliferation [46,47]. Disturbing or blocking the balance of FSH can lead to reduced fertility and even infertility [48,49]. As the receptor for FSH, FSHR is mainly expressed on Sertoli cells [50,51] and also plays a key role in spermatogenesis and Sertoli cell proliferation. In the absence of FSH and FSHR, the number of Sertoli cells in the testis is significantly reduced by 30–45% [52]. Studies in FSHR knockout mice have shown that the absence of FSHR results in sperm abnormalities and a significant reduction in sperm count [53].

The testicular index, sperm density, and sperm motility in the epididymis were significantly lower in pubertal rats immunized with FSHR than those in the control group. However, there was no significant difference in testosterone levels between the two groups. In adult rhesus monkeys immunized with oFSH, the spermatogenesis was inhibited, and testicular atrophy occurred; however, the serum testosterone did not decrease [9]. This is consistent with the results of this experiment. This may be due to the fact that FSH mainly affects the proliferation of Sertoli cells, while testosterone is secreted by Leydig cells [54]; therefore, FSH immunization does not significantly affect the secretion of testosterone [55]. Since FSH plays a vital role in spermatogenesis, FSHR immunization significantly decreased the sperm density and sperm motility. Yan et al. [42,56] screened four B cell epitopes of FSHR and immunized adult male mice with each of them. Three of the four peptides had the effect of inhibiting fertility. The testes of immunized mice atrophied, the serum testosterone level decreased, and the sperm count reduced. Yang LH [41] concatenated the PADRE T cell epitope from Yan’s study with amino acids 32–44 on the FSHR and used this vaccine to immunize mice. Although immunization significantly reduced sperm count and motility and increased deformity rates, it did not lower the serum testosterone levels or cause histological changes in seminiferous tubules and interstitial cells. These results are consistent with the present experiment.

In this study, we prepared recombinant FSHR and LHR subunit vaccines, and both vaccines successfully stimulated the production of corresponding antibodies in prepubertal and pubertal rats. The FSHR vaccine significantly inhibited the development of testes and spermatogenesis in prepubertal and pubertal rats, while the LHR vaccine inhibited the spermatogenesis and testosterone in prepubertal rats only. The overall results of this study suggest that the FSHR subunit vaccine may be a promising approach for immunocastration but still needs improvements in effectiveness.

## Figures and Tables

**Figure 1 vetsci-11-00176-f001:**
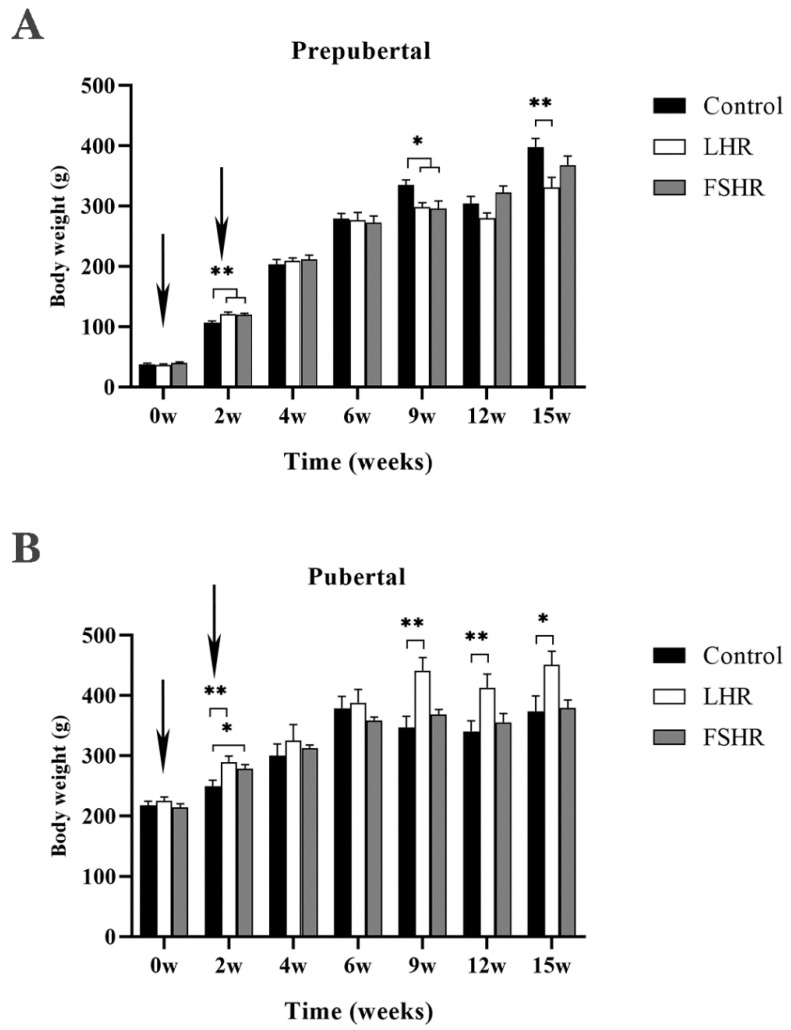
Effects of LHR and FSHR immunizations on body weight in prepubertal and pubertal rats (*n* = 9). Note: Prepubertal rats were immunized for the first time at 20 days of age, and pubertal rats were immunized for the first time at 6–8 weeks of age. All rats were weighed in the afternoon at 0, 2, 4, 6, 9, 12, and 15 weeks after the first immunization. (**A**) shows the changes in body weight of prepubertal rats after vaccination, and (**B**) shows the changes in body weight of pubertal rats after vaccination. * indicates *p* < 0.05, and ** indicates *p* < 0.01. The arrows indicate the time of vaccination.

**Figure 2 vetsci-11-00176-f002:**
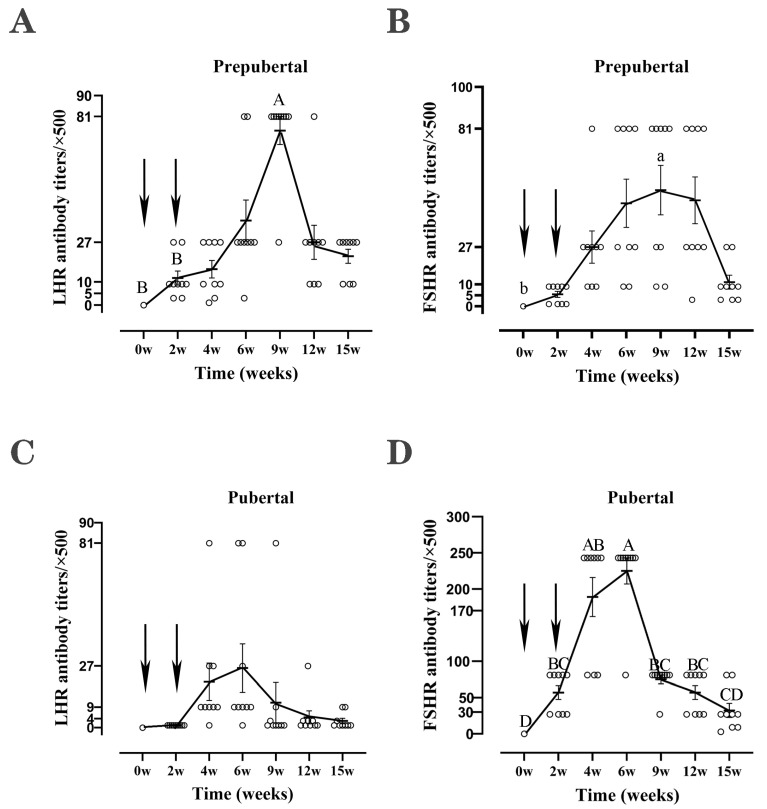
Antibodies titers in prepubertal and pubertal rats immunized with the recombinant LHR or FSHR subunit vaccine (*n* = 9). Note: (**A**,**B**) show the antibodies titers at different time points in prepubertal rats immunized with recombinant LHR and FSHR subunit vaccines, respectively. (**C**,**D**) show the changes in antibody levels in the pubertal rats immunized with recombinant LHR and FSHR subunit vaccines, respectively. Completely different lowercase letters indicate *p* < 0.05; Completely different uppercase letters indicate *p* < 0.01; the arrows indicate the time of vaccination.

**Figure 3 vetsci-11-00176-f003:**
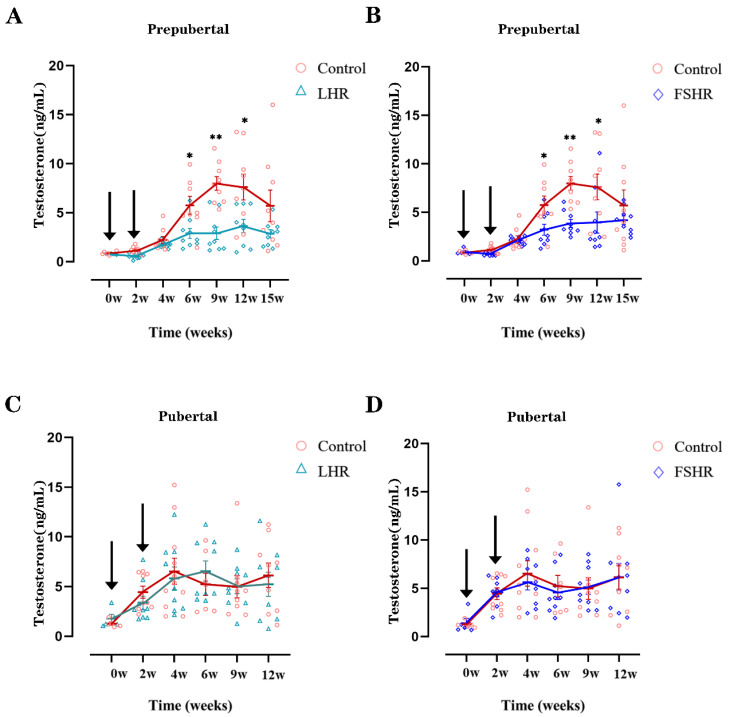
Effects of LHR and FSHR immunizations on serum testosterone levels in prepubertal and pubertal rats (*n* = 9). Note: (**A**,**B**) show the changes in testosterone levels in the prepubertal rats immunized with LHR and FSHR subunit vaccines, respectively. (**C**,**D**) show the changes in testosterone levels in the pubertal rats immunized with LHR and FSHR group subunit vaccines, respectively. The arrows indicate the time of vaccination. * indicates *p* < 0.05. ** indicates *p* < 0.01.

**Figure 4 vetsci-11-00176-f004:**
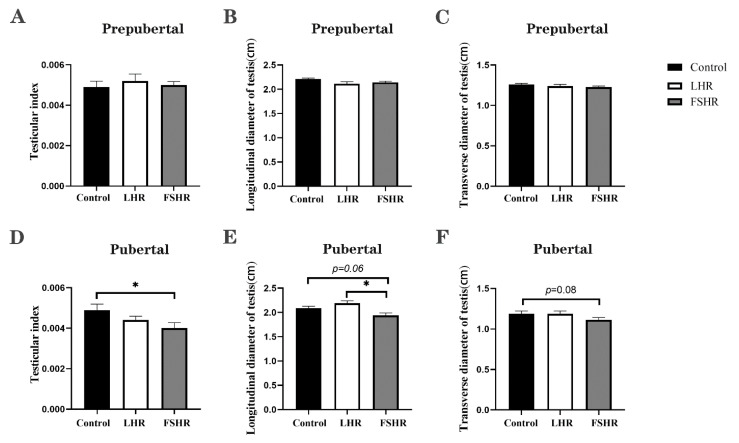
Effect of vaccination on testicular size (*n* = 9). Note: (**A**,**D**) show the testis indices of the prepubertal rats and the pubertal rats, respectively (testis index = average weight of both testicles/body weight × %); (**B**,**E**) show the longitudinal diameter of the testes of the prepubertal rats and the pubertal rats, respectively; (**C**,**F**) show the transverse diameters of testes in prepubertal rats and pubertal rats, respectively. The testicular transverse diameter and testicular longitudinal diameter are both the average values of both testicles. * indicates *p* < 0.05, and no * indicates *p* > 0.05.

**Figure 5 vetsci-11-00176-f005:**
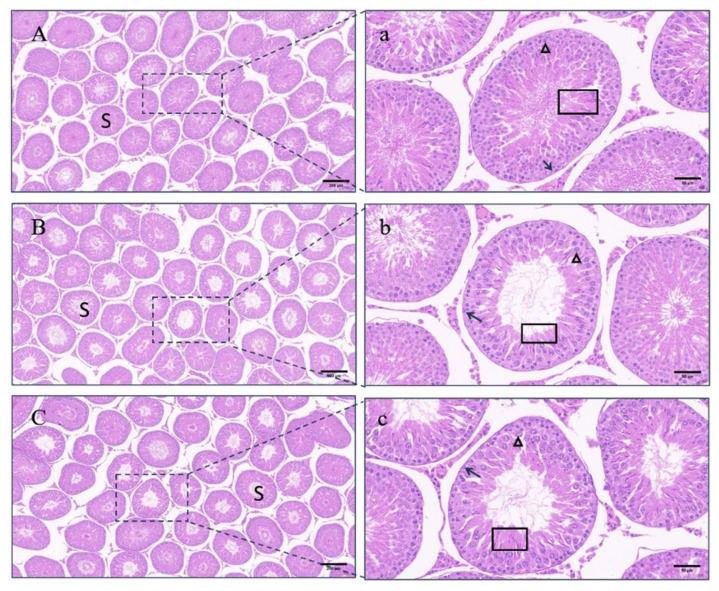
Effects of LHR and FSHR immunizations on the testicular development of prepubertal rats. Note: (**A**) control group (100×); (**B**) LHR group (100×); (**C**) FSHR group (100×); (**a**) control group (400×); (**b**) LHR group (400×); (**c**) FSHR group (400×); S: seminiferous tubules; boxes: spermatozoa; darts: spermatogonia; triangles: primary spermatocytes.

**Figure 6 vetsci-11-00176-f006:**
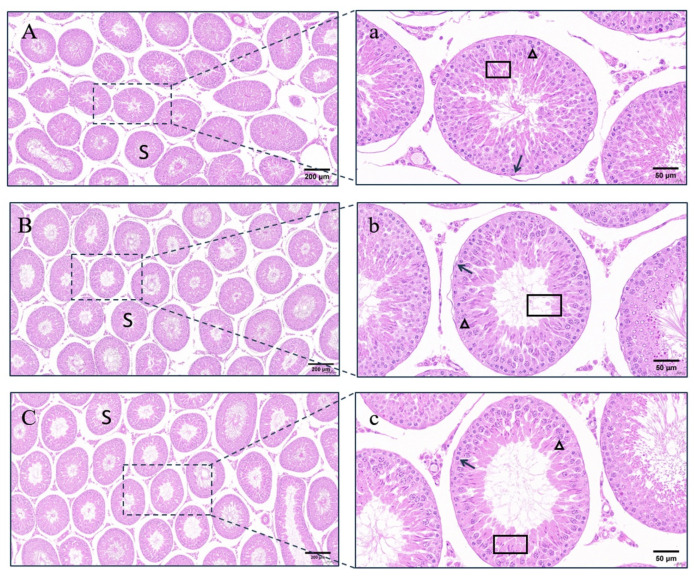
Effects of LHR and FSHR immunizations on the testicular development of pubertal rats. Note: (**A**) control group (100×); (**B**) LHR group (100×); (**C**) FSHR group (100×); (**a**) control group (400×); (**b**) LHR group (400×); (**c**) FSHR group (400×); S: seminiferous tubules; boxes: spermatozoa; darts: spermatogonia; triangles: primary spermatocytes.

**Figure 7 vetsci-11-00176-f007:**
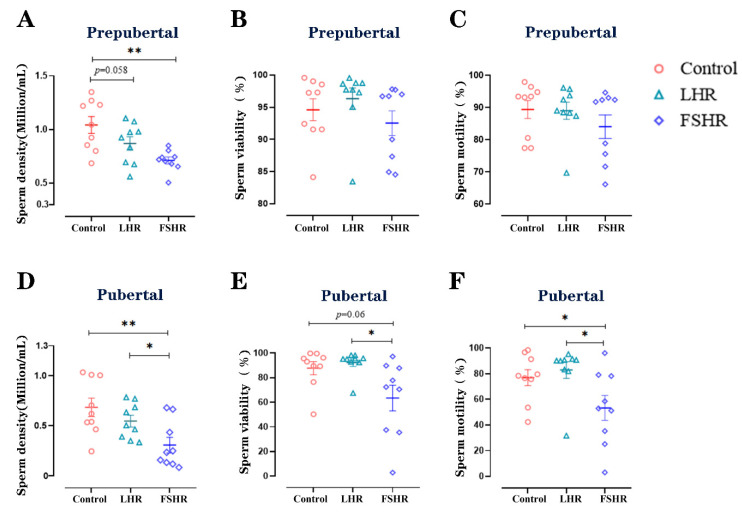
Effects of LHR and FSHR immunizations on the sperm quality in the cauda epididymidis of rats. (*n* = 9). Note: (**A**–**C**) show the effects of LHR and FSHR immunizations on the density, viability, and motility of sperm in the cauda epididymidis of prepubertal rats, respectively. (**D**–**F**) show the effects of LHR and FSHR immunizations on the density, viability, and motility of sperm in the cauda epididymidis of pubertal rats. * indicates *p* < 0.05. ** indicates *p* < 0.01.

## Data Availability

All data in this article are true and reliable, and all experimental models can be repeated. The datasets generated and/or analyzed during the current study are not publicly available [the relevant data are still being applied for a patent] but are available from the corresponding author upon reasonable request.

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
