# Peer review of "Effects of Vaccination against Recombinant FSH or LH Receptor Subunits on Gonadal Development and Functioning Male Rats"

_vetsci, 2024, doi:10.3390/vetsci11040176_

Round 1

Reviewer 1 Report

Comments and Suggestions for Authors

Immunological castration has many advantages compared to surgical castration. In this manuscript, Pan and his colleagues designed two subunit vaccines by fusing conserved antigenic epitopes from pig FSHR and LHR with the T helper epitope region of diphtheria toxin (DTT) and then validated in male rats. The results showed that both vaccines could suppress testicular development and reduce testicular function to some extent. This study is quite interesting. However, authors should answer the following question:

1. Why did authors validate the vaccines using male rats as experimental animals, when the vaccines are intended for future application in livestock such as pigs, dogs, and cats? After all, the immune system and reproductive system development of rats may differ from those of livestocks.

2. Figure 1 demonstrates that immunocastration significantly affects the weight gain of rats. Why did the author not discuss this result?

3. In Figure 4, the protein gel electrophoresis shows that the purified LHR-DTT and FSHR-DTT proteins have impurities. Did the author remove these impurities before immunizing the rats? If the impurities were not removed, could they have an impact on the experimental results?

Author Response

Thank you very much for taking time to review our manuscript and providing a very professional feedback. Your suggestions are very helpful in improving the quality of our manuscript. Here are our responses to your questions:

1. Why did authors validate the vaccines using male rats as experimental animals, when the vaccines are intended for future application in livestock such as pigs, dogs, and cats? After all, the immune system and reproductive system development of rats may differ from those of livestocks.

Answer: Before conducting efficacy in rats, the effect of the two vaccines mentioned in this manuscript were unknown. Considering the cost of the experiment and the controllability of the experimental conditions, we decided to perform preliminary validation in rats first. Based on the results of this experiment, we have optimized the vaccines, and initiated the process of validating in pigs and goats.

2. Figure 1 demonstrates that immunocastration significantly affects the weight gain of rats. Why did the author not discuss this result?

Answer: The focus of this study is mainly on the effects of two vaccines on the reproductive aspects of male rats. When rats were used as research subjects, I personally believed that there was not much discussion value regarding their weight changes. However, the significant impact of immune castration on the weight of immunized rats has been given special attention in our subsequent livestock experiments.

3. In Figure 4, the protein gel electrophoresis shows that the purified LHR-DTT and FSHR-DTT proteins have impurities. Did the author remove these impurities before immunizing the rats? If the impurities were not removed, could they have an impact on the experimental results?

Answer: This question is very professional. Theoretically, impurities could also induced rats to produce the corresponding antibodies. In this experiment, the mount of impurities was much less than the target protein, and the primary goal was to verify whether the two vaccines could produce antibodies against FSHR and LHR, therefore, we did not remove the impurities from the purified proteins before immunizing rats. However, in the subsequent commercialization and improvement of the vaccine, we will pay attention to improving the purity of the protein.

Reviewer 2 Report

Comments and Suggestions for Authors

Review of the manuscript vetsci-2880150 entitled “Effects of two kinds of gonadotropin receptor subunit vaccines on gonadal development in male rats

General comments

This manuscript evaluated the effects of two vaccines, prepared agais LHR and FSHR, on testicular morphophysiology in rats. The main finding was that the FSHR vaccine can be used for immunocastration in prepubertal rats. 

In general, the manuscript is clear and the experimental design appropriated. The authors, however, should be more conservative in their conclusions (e.g., “These two immunocastration vaccines are expected to pave the way for the popularization of immunocastration…”). In the pubertal rats, there was no difference in testosterone concentrations after immunization, which is key to obtain the desired effects of immunocastration on animal stress and aggressiveness. And although there was a difference in average sperm parameters after immunization against FSHR, some individual rats still scored like the controls, and were likely fertile. This is quite different from what we see in experiments with immunization agaist GnRH, in which differences in both testosterone and sperm quantity and quality between treated and control animals are remarkable. The overall results of this study suggests that this may be a promising approach, but still needs improvements in effectiveness. Another point that requires a review is the statistical analysis. In this kind of experiment, in which the effect of treatment on many endpoints occurs progressively over time, the statistical analysis shall use a model to account for the time and time x treatment effects. Other specific comments are listed below.

Specific comments:

Line 2 (title): The title is somewhat confusing. The study did not use two kinds of vaccine, but vaccines against prepared against two gonadotropin receptors. I suggest to change as follows: Effects of vaccination against recombinant FSH or LH receptor subunits on gonadal development and funtion in male rats.

Line 21: both (b lowercase)

Line 23 and throughout text: When the P-value is shown, it is a redundancy to use “significantly”, or “not significant”. E.g.: FSHR-DTT decreased the sperm density […] (< 0.01), …

Avoid starting sentecnes with abbreviations.   

Line 28: “animals” mean that the results can be extrapolated to other species, which is a speculation. Please refer to the species evaluated.

Lines 32-35 and elsewhere: The authors used “immunecastration”, “immune castration” and “immunocastration”. Please standardize as immunocastration 

Line 39: …improve carcass traits…

Lines 48-49: The GnRH I, which is usually referred to as GnRH, regulates gonadotropin secretion.

Lines 63: …antibodies produced by vaccination against FSH or LH…

Lines 74-75: Do the authors have an explanation for such reduction in the interest on LHR/FSHR vaccines over the past years?

Lines 91-92: Is there a protocol number or reference for this approval?

Lines 95-100. Please avoid such long sentences.

Line 102 and elsewhere: please check if the abbreviation for hour is h (lowercase)

Line 109 and other Figure Notes: Please fuse these notes with the figure captions. 

Line 114 and throughout text: Delete “extremely”. The significance is shown by the P-value.

Line 124: Avoid starting sentecnes with numbers. Rewrite as follows: Prepubertal (20 days old, n=30) and pubertal (6~8 weeks old, n=30) male rats…

Line 125: …randomly allocated…

Line 142: Then the plate was sealed with…

Lines 142-147: Verbs are in the present tense (add, wash, incubate). Please rewrite accordingly

Line 152 (Figure 3): In both cases (LHR and FSHR), testosterone seems to decrease over time from week 9 on. How do the authors explain this?

Line 175: Euthanized, not killed

Line 182: …China?

Line 190: The authors should have used an statistical model that accounted for the effects of treatment, time, and treatment x time interaction. I recommend to review the statistical analysis.

Line 196: To test the effect of immunization against FSHR or LFR on…

Line 202: Avoid superlatives such as “dramatically”. The difference among groups is noticeable, but far from “dramatic”. 

Line 209: The ATB titers differed statistically among days (i.e., was there a effect of time)?

Line 259: … viability and motility were evaluated.

Lines 261-262: “… and the sperm density in LHR group was also lower […], but there was no significant difference…” If there was no statistical difference, do not state that it “was also lower”.

Line 296: … the secretion of testosterone was inhibited…

Comments on the Quality of English Language

The text is in general well written, but a general English review is recommended, particularly for verb tenses and wording

Author Response

Thank you very much for taking time to review our manuscript and providing a very professional feedback. Your suggestions are very helpful in improving the quality of our manuscript. Here are our responses to your questions:

1. Lines 74-75: Do the authors have an explanation for such reduction in the interest on LHR/FSHR vaccines over the past years?

Answer: Compared to immunocastration vaccines using GnRH or kisspeptin as immunogens, the effectiveness of using terminal hormones/receptors such as FSHR and LHR as immunogenic vaccines is indeed not ideal. Due to the cascade amplification effect in hormone regulation, immunocastration vaccines targeting upstream hormones have indeed shown more significant effects, which may be the reason for the decrease in interest in LHR/FSHR vaccines in recent years. However, compared to GnRH, FSHR and LHR are lower in the reproductive axis and more targeted, directly regulating reproductive functions at the gonadal level. This means that immunointervention targeting these receptors can more directly impact the development and function of reproductive cells, with less disruption to the upstream hormone axis. On the other hand, compared to continuously secreted hormones, receptors have a slower turnover rate; thus, targeting receptors could lead to higher and more prolonged antibody titers. Therefore, it is necessary to continue improving FSHR/LHR vaccines.

2. Line 152 (Figure 3): In both cases (LHR and FSHR), testosterone seems to decrease over time from week 9 on. How do the authors explain this?

Answer: Starting from the 9th week, the testosterone levels in the control group of prepubertal rats began to decrease. It is speculated that the factors other than experimental factors caused stress, which leads to a decrease in testosterone levels. For example: The ambient temperature is too hot and the air conditioner is used for a long time, the temperature regulation effect of the air conditioner cannot be maintained at 23-26 ℃, resulting in heat stress. Alternatively, at the beginning of the experiment, the prepubertal rats may have a smaller body size. As the size and weight of the rats increase, the space originally occupied by three rats in one cage has become crowded, leading to stress in the rats. However, due to their larger size at the beginning of the experiment, Pubertal rats were always kept in a cage of two, so there was no stress. For the above two possibilities, we are only making a speculation, and the specific reasons still need further exploration.

3. Line 190: The authors should have used an statistical model that accounted for the effects of treatment, time, and treatment x time interaction. I recommend to review the statistical analysis

Answer: We used one-way repeated measures ANOVA to explore the relationship between time and antibody levels, and replaced the original Figure 2 with a new one.

Reviewer 3 Report

Comments and Suggestions for Authors

The present manuscript reports on vaccinating against FSHR or LHR on gonadal development in male rats. They studied vaccinations in prepubertal and post-pubertal rats. Both vaccines reduced testicular function and sperm production. However, the FSHR vaccine was more effective, leading to the conclusion that it would be promising for use in prepubertal animals.

Figure 1 Why are the prepubertal animals heavier (~275 gm) at 6-9 weeks than the pubertal animals @ O weeks when the first immunization is at 6-8 weeks of age (~200 gm)?

                Indicate n in the legend for each group.

                Differences in body weight results from vaccines?

How are these body weight data used to test a hypothesis about vaccination effects?

L 188     Were p values adjusted for multiple comparisons to the same control? Apparently, you have used more degrees of freedom than you should have without adjusting the p-value.

Comments on the Quality of English Language

L 14        rephrase - “play key roles”

L 18        Sprague Dawley (SD)

L 39        rephrase - Castration is usually practiced to reduce…

L 51        delete “till nowadays”

L 53        change to cause

L 56        rephrase - “play crucial roles”

L 73        delete “proven” delete “the”

L 75        delete “currently”

L 87        not clear meaning “of the same family”

L 131     here and throughout change sperms to sperm

L 141-148           add helping verbs - the plate was sealed, the serum was added, was incubated, was washed, was added to, etc.

L 185     rephrase - and was stained with hematoxylin and eosin

L 199     rephrase – “). The body” makes two sentences

L 221     delete sentence

L 259     delete experiment

L 263     sperm

L 298     rephrase – “In contrast”

L 315     rephrase – “In adult rhesus monkeys immunized with…”

L 320     delete so

L 324     count was lower

L 418     capitalize O

L 504     lower case title words after Role

Author Response

Thank you very much for taking time to review our manuscript and providing a very professional feedback. Your suggestions are very helpful in improving the quality of our manuscript. Here are our responses to your questions:

1. Figure 1 Why are the prepubertal animals heavier (~275 gm) at 6-9 weeks than the pubertal animals O weeks when the first immunization is at 6-8 weeks of age (~200 gm)?

Answer: Prepubertal rats were 20 days old at the time of initial immunization, and at 6-9 weeks after initial immunization, their actual age was 9-12 weeks old  (~275g). Pubertal rats were first immunized at 6-8 weeks of age. Prepubertal rats at 6-8 weeks old corresponded to 3-5 weeks after the first immunization, with a weight of approximately 200 grams, which is similar to the weight of pubertal rats at 6-8 weeks old.

2. Differences in body weight results from vaccines? How are these body weight data used to test a hypothesis about vaccination effects?

Answer: From the results of this experiment, it can be found that vaccine immunity does  indeed affect the weight change of rats to a certain extent, but the focus of this article is mainly on the impact of vaccines on the reproductive system of rats, so there is no discussion on the weight change. However, the impact of vaccines on weight has also attracted our attention. We have optimized and improved the vaccine based on this experiment, and have focused on exploring weight changes in subsequent experiments on meat animals such as goats and pigs.

3. L188 Were p values adjusted for multiple comparisons to the same control? Apparently, you have used more degrees of freedom than you should have without adjusting the p-value.

Answer: When there are three or more groups, we use One-Way ANOVA, and when comparing two groups, we use independent t-test. It has been specifically marked in the article.
